# Immunologic Aspects in Fibrodysplasia Ossificans Progressiva

**DOI:** 10.3390/biom14030357

**Published:** 2024-03-16

**Authors:** Anastasia Diolintzi, Mst Shaela Pervin, Edward C. Hsiao

**Affiliations:** Division of Endocrinology and Metabolism, Department of Medicine, the Institute for Human Genetics, the Program in Craniofacial Biology, University of California, San Francisco, CA 94143, USA; anastasia.diolintzi@ucsf.edu (A.D.); mstshaela.pervin@ucsf.edu (M.S.P.)

**Keywords:** inflammation, macrophages, immune activation, cytokines, fibrodysplasia ossificans progressiva (FOP), heterotopic ossification

## Abstract

Background: Inflammation is a major driver of heterotopic ossification (HO), a condition of abnormal bone growth in a site that is not normally mineralized. Purpose of review: This review will examine recent findings on the roles of inflammation and the immune system in fibrodysplasia ossificans progressiva (FOP). FOP is a genetic condition of aggressive and progressive HO formation. We also examine how inflammation may be a valuable target for the treatment of HO. Rationale/Recent findings: Multiple lines of evidence indicate a key role for the immune system in driving FOP pathogenesis. Critical cell types include macrophages, mast cells, and adaptive immune cells, working through hypoxia signaling pathways, stem cell differentiation signaling pathways, vascular regulatory pathways, and inflammatory cytokines. In addition, recent clinical reports suggest a potential role for immune modulators in the management of FOP. Future perspectives: The central role of inflammatory mediators in HO suggests that the immune system may be a common target for blocking HO in both FOP and non-genetic forms of HO. Future research focusing on the identification of novel inflammatory targets will help support the testing of potential therapies for FOP and other related conditions.

## 1. Introduction

Bone homeostasis is closely linked with immune function. The key cell types responsible for maintaining the balance between bone formation and resorption are osteoblasts and osteoclasts. Osteoblasts originate from mesenchymal stem cells (MSCs) in the long bones, or MSC-like cells in the craniofacial bones, specifically in the neural crest lineage [1]. Conversely, osteoclasts originate from hematopoietic progenitors from the monocyte/macrophage lineage [2]. Osteoclast formation is driven by the binding of the receptor activator of Nuclear factor kappa-light-chain-enhancer of activated B cells (NF-*κ*B) ligand (RANKL) to the RANK receptor [3,4]. Additionally, macrophages appear to promote osteoblast formation, as evidenced by studies showing that macrophage-deficient mice exhibit reduced bone density and impaired MSC differentiation into osteoblasts [5]. Osteal macrophages (also known as “osteomacs”, a specific subtype of macrophages residing in bone tissues) collaborate with pro-inflammatory macrophages to stimulate anabolic bone formation during fracture repair [6]. Furthermore, various diseases affecting the immune system or causing autoimmunity are associated with bone loss [7]. Together, these findings suggest that innate immune cells, including macrophages, may play a significant role in the development of bone disorders and could potentially serve as therapeutic targets for the treatment of a spectrum of bone diseases.

While inflammation is essential for normal tissue repair, an uncontrolled inflammatory response can lead to pathological conditions [8]. The connections between the immune system and conditions of abnormal bone formation, such as heterotopic ossification (HO), have been investigated for several decades. HO is a severe and debilitating process of abnormal bone formation that occurs at non-mineralized sites [9,10]. HO can be triggered by trauma or burn injuries, or in conditions of severe inflammation such as in rheumatologic diseases [10,11,12,13,14].

## 2. Inflammatory Responses in the Context of Bone

Inflammatory cytokines and inflammatory cells are considered major drivers of HO formation after injuries or in pathologies prone to HO [15,16,17,18,19,20]. Pro-inflammatory cytokines, such as interleukin 3 (IL-3), are elevated in patients with combat wounds [18]. Additionally, in murine models, interleukin (IL) 1β (IL-1β) secreted by the inflammasome is elevated in bone marrow-derived macrophages (BMDMs), while suppression of IL-1β secretion can cause recovery of bone mass in ovariectomized mice [21,22]. Interestingly, in human MSCS, inflammasome stimulation results in enhanced adipogenesis and decreased osteogenesis [21,23]. Another member of the interleukin family, interleukin 6 (IL-6), activates bone degradation, suggesting potential involvement in osteoporosis [24]. The natural antagonist of the pro-inflammatory cytokine IL-18, IL-18BP, has been found to be low in osteoporotic women; animal studies indicate a stimulatory role of IL-18BP in osteoblast differentiation, preservation of cortical bone, and restoration of trabecular microarchitecture, as well as an inhibitory role in inflammasome activation and osteoclastogenesis [25]. Further, loss of the anti-inflammatory cytokine IL-10 worsens the bone loss in type 1 diabetes [26]. Another bone-protecting cytokine is IL-33, which may hold therapeutic potential against bone resorption [21]. Additionally, TNF-α is known to repress osteoblast function during differentiation, while triggering osteoclast proliferation and differentiation [27]. Non-genetic HO after high energy trauma in penetrating war injuries show elevations in IL-6, IL-10, and monocyte chemoattractant protein 1 (MCP-1) [17]. Based on the large number of inflammatory cytokines found in HO, a number of anti-inflammatory therapies are used clinically in an attempt to decrease the risk of HO after major hip procedures, although the blockade is not complete [16,19,28].

Because of the heterogeneity of triggers in non-genetic HO, it is useful to examine the signaling pathways and mechanisms active in genetic forms of HO. Much has changed since our last review of this area in 2019 [29]. This review updates the current state of knowledge regarding macrophages’ involvement in fibrodysplasia ossificans progressiva (FOP), a genetic disorder of severe progressive HO formation. We also emphasize the impact of the causative activin A type 1 receptor (ACVR1) genetic mutations on the inflammatory microenvironment, fibrotic tissue transformation, and unique interactions with other signaling pathways. Additionally, we discuss potential therapeutic strategies targeting the immune system being studied for FOP management.

## 3. Molecular Aspects of Osteogenic and Chondrogenic Differentiation

Bone remodeling comprises a series of sequential events [30]. The first is activation, where elusive stimulatory triggers determine the starting point of bone formation by activating osteoblasts and bone lining cells. This is the location to which osteoclasts are then summoned and where they transform into their mature form [31]. Next, the previously developed mature osteoclasts enzymatically degrade the inorganic matrix during the resorption phase. After bone resorption is complete, osteoclast apoptosis follows, and macrophage-like cells drift to the resorbed area to remove any remaining osteoclast-generated debris in a step called reversal. During the end stages of the reversal process, when the osteoclast-mediated bone resorption wanes, osteoclasts secrete sphingosine 1-phosphate, which incites osteoblast recruitment [32]. Next, more osteoblasts are recruited to the resorbed site, where they assemble a new inorganic matrix, and the subsequent mineralization fills up the resorption lacuna during formation. After formation, the bone remodeling cycle concludes with the termination phase, whereby osteoblasts have three possible fates, meaning that they undergo apoptosis, transform into bone lining cells, or become trapped into the bone matrix, where they end up differentiating into osteocytes [30,32]. The bone matrix-embedded osteocytes will control and coordinate future events of bone remodeling [32].

Bone tissue turnover and regeneration require crucial, yet complex and not fully elucidated, interactions with the immune system. Disruption can result in a wide range of inflammatory disorders in the skeleton [30,33]. Intriguingly, there is a crosstalk between cells of the immune system and cytokines, with bone cells (i.e., osteoblasts, osteoclasts, and osteocytes) implicated in the modulation of both bone homeostasis and immune cell proliferation, differentiation, and function in target cells. This bidirectional crosstalk has been termed ‘osteoimmunology’ [30,32,34,35]. Immune cells involved in bone regeneration include neutrophils, macrophages, dendritic cells, innate lymphoid cells, T cells, Th1/Th2 and Th17 cells, T regulatory cells (Tregs), and B cells [30,35]. Additionally, bone cells and immune cells are not merely co-present in the bone marrow milieu, but also appear to have overlapping progenitor cells and modulatory molecules, which renders their two-way communication more intricate [30].

Immune cytokines that play an essential role in bone regeneration include TNF-α, IL-1, IL-6, IL-17, and IFN-γ, as well as numerous other cytokines (e.g., IL-12, IL-23, IL-27, IL-35) [30,33,36]. More specifically, the immune-related cytokines IL-1β, ΙL-6, IL-17, and TNF-α are thought to be the primary inducers of osteoclastogenesis by enhancing the expression of factors involved in osteoclast differentiation, such as RANKL and the macrophage-colony-stimulating factor (M-CSF), favoring bone resorption [33,37]. Another class of inflammatory molecules promoting osteoclastogenesis is that of the chemokines, involving several C-C motif chemokine ligands (CCLs) (e.g., CCL3, CCL5, CCL9, etc.), the C-X3-C motif chemokine ligand (CX3CL1), as well as the X-C motif ligand 1 (XCL1), all of which enhance osteoclast differentiation [33,37]. Some of these inflammatory molecules exert adverse effects on osteoblasts by inhibiting their differentiation and stimulating osteocyte apoptosis, leading to decreased formation, mineralization, and density in osseous tissue [33]. In contrast, other immune- and bone-cell-secreted factors, such as osteoprotegerin (OPG), which functions as a decoy receptor for RANKL, impede osteoclast differentiation [31,33], while immune-related molecules, such as IL-4, IL-10, IL-27, interferons, annexin A1, and cardiotrophin-like cytokine factor 1 (CLCF1), constrain osteoclastogenesis and osteoclast differentiation [33].

Bone remodeling is primarily maintained by the absorptive properties of osteoclasts and the bone formative properties of osteoblasts in a finely tuned spatiotemporal manner, with the additional contribution of osteocytes that sense mechanical stressors, and bone lining cells that instigate skeletal turnover, by degrading the bone matrix [30,31,32,35,36], and capillary blood supply [32]. The bone forming osteoblasts arise from multi-potent MSCs and their main actions are to generate bone matrix proteins and mineralize the bone, but which also express factors involved in osteoclastogenesis [32]. Together, these different types of cells form an intricate complex termed a ‘basic multicellular unit’ (BMU), which is essential for efficient bone remodeling [32]. The transcription factor runt-related transcription factor 2 (RUNX2) is indispensable for osteoblast development and differentiation [32,35]. Separately, osteoclasts emanate from the monocyte/macrophage hematopoietic lineage and hold the capacity to degrade mineralized bone, cartilage, and dentine [31,32]. Osteoclast maturation and function are also necessary, and can be followed by osteoclast markers (e.g., integrin receptors, cathepsin K, calcitonin receptor, tartrate-resistant acid phosphatase), RANKL and M-CSF [31,32,35].

## 4. Inflammatory Responses and Inflammation-Associated Molecules and Immune Cells

The contribution of the immune and cytokine microenvironment to skeleton turnover and regeneration is critical for bone homeostasis and remodeling across the lifespan. For example, activated macrophages help direct the differentiation of osteoblasts, possibly through the secretion of oncostatin M (OSM) [38]. Additionally, the anabolic properties of parathyroid hormone on bone (i.e., IL-6 and IL-6 receptor (IL-6R)) may be mediated by T cells [39]. Interestingly, defects in the calcineurin (CN)/nuclear factor of activated T cells (NFAT) signaling pathway have been implicated in the enhancement of lipopolysaccharide (LPS) tolerance in macrophages, activating T cells and natural killer cells, and regulating B cell development—all of which may affect osteoclastogenesis regulation [32]. The wingless-related integration site 1 (Wnt1) has recently been shown to be produced by B-cells within the bone marrow niche, and to induce bone formation [30,35,40]. Injury of bone tissue stimulates an early innate immune response consisting of neutrophils, mast cells, monocytes, and macrophages, whereas a later adaptive immune response also occurs, consisting of T and B cells [30].

Infection is another trigger for the initiation of complex cell–cell communication between osteoclasts, osteoblasts, and immune cells [32]. Bone resorption during infection is modulated by both the innate and adaptive immune system, with the involvement of T cells, Tregs (i.e., specialized CD4 T cell subgroup), Th17 cells (i.e., IL-17-producing helper T cell), B cells, neutrophils, dendritic cells, and macrophages [32]. Immune cells serve as both anabolic and catabolic mediators in bone tissue: in the early stages of bone fracture healing, acute inflammatory response leads to chemokine secretion, which triggers the proliferation of MSCs and their downstream differentiation into osteoblasts, contributing to bone regeneration (i.e., anabolic function) [30]. In contrast, chronic inflammatory response results in bone resorption (i.e., catabolic function) through the repression of osteoblast-stimulated bone formation and the induction of osteolysis [30].

## 5. Fibrodysplasia Ossificans Progressiva Can Be Considered a Genetic Disease of Inflammation-Driven Heterotopic Ossification

FOP is a rare genetic disorder affecting approximately one in 1.36 million to 2 million individuals [41,42]. Clinically, FOP is characterized by large amounts of abnormal bone formation in tissues that are not normally mineralized, such as skeletal muscle, tendons, and connective tissues. Over time, the new HO leads to immobility and severe pain from nerve compression and stretching. Treatments remain largely symptomatic with only one approved directed therapy available.

The main cause of FOP is the presence of an activating mutation in the ACVR1, which is part of the bone morphogenetic protein (BMP) pathway [43]. The identified mutations in FOP are primarily located in the glycine-serine (GS) region of ACVR1, with the most common being the single-point mutation ACVR1 c. G617A/p.R206H [43]. This mutation reduces the stability of the GS region, resulting in continuous activation of ACVR1 and neoligand activity to Activin A, leading to HO and eventual joint fusion in patients with FOP (Figure 1). Other rare missense mutations in the GS or protein kinase (PK) domain of ACVR1 have also been reported in patients with FOP [44,45,46,47]. All of these gene mutations in FOP disrupt normal homeostasis and cell differentiation processes, triggering the abnormal endochondral ossification that is a hallmark of FOP [48].

Activin A serves as a competitive inhibitor of the BMP signaling pathway, as indicated by various studies [52,53,54,55]. Under normal conditions, activin A binds to the wild-type ACVR1 receptor and activates the signaling cascade through the SMAD2/3 pathway. The presence of the ACVR1^R206H^ mutation changes the ACVR1 response to activin A, instead allowing signaling through the SMAD1/5/9 pathway—effectively misinterpreting activin A as a BMP. This is discussed in more detail below.

A striking and pathognomonic clinical feature of FOP is that patients can develop massive inflammatory lesions that ultimately lead to HO formation. These can occur with severe traumatic events but have been reported even after mild trauma [41]. These inflammatory lesions, often called “flare-ups,” are accompanied by classical symptoms of inflammation such as induration, erythema, and pain [56]. Tissue injury exacerbates the progression of FOP development. Histology shows the presence of many cell types in these developing FOP lesions, including macrophages, mast cells, MSCs, osteocytes, chondrocytes, and fibroblasts. Cytokines are reportedly elevated in the connective tissues, and within blood vessels and skeletal muscles [57,58,59,60]. 20% or more of these flare-ups result in significant bone formation. The mainstays of current FOP therapy are anti-inflammatory agents, including nonsteroidal anti-inflammatory drugs (NSAIDs) and corticosteroids. These can help reduce FOP HO formation [28], and are also used in the treatment of non-genetic HO [29]. However, these strategies still show poor efficacy when blocking new HO formation.

## 6. The Role of the Immune System in Fibrodysplasia Ossificans Progressiva

The role of the immune system in FOP has garnered significant interest. Hematopoietic-lineage cells have long been recognized as contributors to the early inflammatory phases [61]. A mouse model with an *ACVR1^R206H^* mutation showed the strong presence of macrophages and mast cells at the site of HO formation [62]. Suppression of the transforming growth factor-beta (TGF-β) in FOP mouse models attenuates HO [63], sugesting that macrophages have a critical role in the early phase of inflammation in developing FOP lesions. Our own research has revealed a significant increase in different cytokine levels, such as IL-3, IL-7, and IL-8 in the blood of patients with FOP [20] and has also shown that NF-*κ*B activation is a key factor for inflammation in FOP. Additionally, we have found that blood samples obtained from FOP individuals have high levels of proinflammatory interleukins even in the absence of flare-up symptoms [29,64]. We also found increased monocyte production of TGF-β in patients with FOP, consistent with findings in mouse models [63]. However, the mechanism by which the ACVR1 mutation induces macrophage hyper-responsiveness remains largely unknown.

## 7. Macrophages in Fibrodysplasia Ossificans Progressiva: A Key Player in Disease Pathogenesis

Most forms of HO formation occur through a process of endochondral ossification [65,66,67]. This process involves four stages: inflammation, cartilage formation, osteogenesis, and ectopic bone maturation. In the inflammatory phase, immune cells infiltrate the injury site, as has been well demonstrated in FOP [68]. As the HO formation process progresses, local MSCs differentiate into chondrocytes [66], followed by chondrocyte hypertrophy and calcification during osteogenesis [67]. In the final stages of HO formation, the bone matures to form cancellous bone [65,66,67].

Although the specific influence of individual macrophage subtypes on chondrogenic formation in HO patients is not yet fully understood, insights can be drawn from other diseases that affect cartilage. Inflammatory factors like TGF-β can promote chondrogenesis and osteogenesis [63]. Studies on osteoarthritis patients suggest that M1 (inflammatory) macrophages can also induce cartilage apoptosis, while M2 (tissue repair) macrophages can promote cartilage hypertrophy and ectopic bone formation [69,70].

Notably, the osteogenic and chondrogenic differentiation of MSCs or MSC-like cells is a critical process in the development of HO [71]. Different subtypes of macrophages are able to regulate the osteogenic differentiation of MSCs [72], and the depletion of macrophages can decrease the osteogenic potential of MSCs [5]. Conversely, MSCs release signaling factors, such as prostaglandin E-2 (PGE-2), that control the polarization of macrophages [73]. Thus, the control induced by different subtypes of macrophages via production of factors like TGF-β1, BMP, activin A, OSM, substance P (SP), and neurotrophin 3 (NT-3), play vital roles in the development of HO by helping to regulate the inflammatory process, bone repair process, and potential progression of HO [57,74,75,76,77,78]

Macrophage depletion can mitigate new HO formation in mice [57]. To overcome the barrier of obtaining large numbers of characterized human cells for understanding the tissue effects of macrophages, several protocols with which to generate macrophages from mouse- [79,80] and human- [51,80,81,82,83,84] induced pluripotent stem cells (iPSCs) have been described. Using methods to create human iPSC-derived macrophages (iMACs) from FOP and non-FOP [51], we have previously shown the heightened production of pro-inflammatory cytokines in unstimulated FOP-M1-like iMACs. This aligns with our earlier observations of a pro-inflammatory state as identified in FOP serum and in primary monocyte-derived M1-like macrophages [20]. Furthermore, the significantly elevated IP-10 (CXCL10) and RANTES (CCL5) in FOP-M2-like iMACs mirrored our observations in FOP-patient-derived primary monocytes. Unexpectedly, cytokine production in M1-like-iMACs reached saturation upon stimulation, with the lowest concentration of LPS that we tested. In contrast, FOP-M1-like iMACs exhibited enhanced inflammatory cytokine production, with stronger differences emerging after the M1-like polarization steps. This disparity in timing indicates that FOP-iMACs may retain a pro-inflammatory phenotype after LPS stimulation, for an extended duration, as compared with control cells. These findings align with our previous experiments involving primary FOP samples, indicating that LPS stimulation of naïve FOP monocytes exhibits prolonged NF-*κ*B activation, compared with control monocytes.

Surprisingly, we found no significant differences in cytokine production responses to the damage-associated molecular patterns (DAMPs) that may be released after tissue injury, suggesting that the initiation process of the macrophages by tissue injury is not changed, but that the magnitude and/or duration is prolonged in FOP [51]. FOP-M1-like macrophages also exhibited increased activin A production, potentially acting as a source of the neo-ligand for the ACVR1^R206H^ receptor. This FOP iMA̧C model provides important opportunities for detailed studies on the way in which *ACVR1* regulates inflammation in the human FOP disease state and highlights the crucial role of macrophages as a driver of HO formation in FOP patients.

## 8. Other Immune Cell Types in Fibrodysplasia Ossificans Progressiva

In addition to myeloid cells, other types of immune cells, such as mast cells and lymphoid cells, are thought to contribute to the inflammation of FOP.

***Mast Cells:*** Mast cells are immune cells known for their involvement in allergic and inflammatory responses [85,86]. These mast cells release pro-inflammatory mediators in patients with FOP and contribute to the inflammatory micro-environment. Depletion of mast cells can reduce HO volume by about 50% in the conditional knock-in mouse *ACVR1^R206H^* model. Additionally, the combined depletion of mast cells and macrophages, together with the use of clodronate, reduced HO volume by about 75%, suggesting that these two cell types are contributors to HO formation in FOP [57]. Our own prior study provides additional support for this concept, as serum isolated from patients with FOP had high levels of interleukin 9 (IL-9), a cytokine produced by mast cells [20].

***Lymphocytes:*** Both B and T lymphocytes have been identified as potential contributors to the development of FOP. Perivascular lymphocytic infiltration has been reported in the skeletal muscle in a 2-year-old child with FOP, despite normal histology [87]. The presence of perivascular lymphocytic infiltrates in FOP lesions suggests that lymphocytes play a role in the inflammatory process. Studies undertaken on animal models have shown that the adaptive immune system may impact the advancement of HO. RAG1 null mice in an Nse-BMP4 background. These mice lack mature B or T lymphocytes, and form HO after injury without any temporal delay. Notably, the loss of adaptive cells decreased the rate of spreading and the overall amount of HO [88]. These findings suggest that adaptive immunity may not be necessary for the initiation of HO but might be a critical regulator for the expansion of HO lesions [88,89]. Prior studies have also shown that HO expansion after a burn injury is attenuated when mature B- and T-lymphocytes are not present [90]. Finally, histology on early FOP lesions has shown extensive perivascular inflammatory infiltrates with a significant expression of hypoxia-inducible factor 1 alpha (HIF1α) [91]. Taken together, these human and mouse data indicate that a complex inflammatory response involving lymphocytes, mast cells, macrophages, and likely other immune cell types are key drivers of FOP HO formation.

## 9. Activin A—A Complex Contributor to Fibrodysplasia Ossificans Progressiva Inflammation

Activin A has an obligatory role in the initiation of FOP HO, through aberrant signaling via the neo-ligand activity of the ACVR1^R206H^ receptor [52,54,92]. However, there are still questions about the precise origin of activin A in FOP patients.

***Activin A as an inflammatory mediator:*** Activin A is a member of the TGF-β/BMP family of ligands [93]. Recently, de Ruiter et al. discovered that TGF-β1 stimulates the generation of activin A in dermal fibroblasts obtained from FOP patients, as evidenced by an increase in the inhibin subunit beta A (*INHBA*) gene and protein expression levels, functioning as an upstream activator of activin A generation, specifically in FOP [94]. These findings suggest that TGF-β may be an additional target along with activin A, as its blockade could hinder, at least in part, the signaling cascade concluding in the HO formation arbitrated by activin A in FOP [94]. This is in agreement with previous reports implicating the TGF-β pathway as a strong driver of HO, as HO has been shown to be abated following the circulating repression of TGF-β in FOP mouse models [63]. Increased inflammatory responses and canonical BMP receptor (BMPR) signaling all occur during the early stages of trauma-induced HO formation in FOP, and it has been proposed that the crosstalk of several inflammatory signaling pathways may exert a synergistic effect when enhancing inducement of Smad1/5/8 activity in various cell types mediated by BMP [95].

***Receptors downstream of Activin A:*** Many of the inflammatory signals induced early after injury are mediated through toll-like receptors (TLRs) and IL-1 receptors via MyD88-dependent pathways [95]. Despite the reported synergistic effect of IL-1β and LPS when the phosphorylation of Smad1/5 is enhanced by activin A in *ACVR1^R206H^*-expressing fibroadipo-progenitors (FAPs), the adapter protein MyD88 has been indicated to be dispensable for trauma-induced HO formation in FOP [95]. Further, it has been demonstrated that activin A promotes early differential gene expression in ligament fibroblasts from FOP patients, with the activin, TGFβ, and BMP signaling being significantly enriched, and upregulated transcript levels in genes involved in bone metabolism (SHOC2 Leucine Rich Repeat Scaffold Protein (*SHOC2)*, Tetratricopeptide Repeat Domain 1 (*TTC1)*, Tetratricopeptide Repeat Domain 1 (*PAPSS2)*, Dedicator Of Cytokinesis 7 (*DOCK7)*, and Lysyl Oxidase (*LOX*)) [96]. Activation of ACVR1^R206H^ by activin A can also occur by receptor-dependent clustering, thus inducing its auto-activation without the need for upstream kinases that are otherwise required for WT ACVR1 activation. This appeared to employ the type II receptors ACVR2A/B instead [97].

Surprisingly, ACVR1 is the only subtype of the BMP type I receptors capable of interceding in the activin A-induced BMP signaling that is consequent to the disengagement of FKBP12, while FKBP12 inhibition results in failure of BMP4 to cross-activate the TGF-β pathway [98]. Additionally, type II BMP receptors can enhance activin A-induced BMP signaling via their kinase activity [98]. Recently, polypeptide substrate accessibility has been proposed as another potential mechanism, where the ACVR1 gain-of-function R206H mutation allosterically changes the ACVR1 kinase activity [99] and results in altered responsiveness to activin A.

***Pathways potentially involved in FOP heterotopic bone formation:*** Intriguingly, FOP HO development has been shown to be closely correlated with spatiotemporal patterns of human body infrared thermographs, suggesting that temperature responsive FOP connective tissue progenitor cells (CTPCs) may act as potent determinants of the anatomic distribution of HO in FOP [100]. Recently, vitamin D_3_ has been raised as another potential modulator of BMP signaling, chondrogenesis, and possibly enhanced activin A expression. This is an intriguing area of study as it may highlight the importance of lifestyle factors, such as diet, as potential disease modifying factors in the context of FOP [101].

Another pathway likely involved in aberrant chondrogenesis in FOP implicates the mammalian target of rapamycin (mTOR) signaling [102]. By employing a high-throughput screening (HTS) using FOP-patient-derived iPSCs, Hino and colleagues found that the lysophosphatidic acid-producing enzyme ectonucleotide pyrophosphatase/phosphodiesterase 2 (ENPP2) may link ACVR1^R206H^ and mTOR signaling in chondrogenesis via an activin A/ACVR1^R206H^/ENPP2/mTOR axis [102].

As noted previously, the pathways involved in inflammation in FOP show significant roles for IL-1 and p38 [20,102]. Elevated IL-1β plasma levels have been found in a case study of FOP patient with repeated and extremely active flare-ups, which were mitigated with anti-IL-1 treatment, as evidenced by reported flare-up occurrence and IL-1β levels [103]. In marked contrast, no differences in systemic activin A levels have been detected between patients with FOP and healthy individuals, even when a patient with FOP has an active flare-up [104]. This finding has led to the hypothesis that activin A levels may be regulated at the local level, highlighting the pivotal role of the local environment in driving HO formation in FOP. Nonetheless, further research is warranted in order to elucidate the source of these pro-inflammatory cytokines, as well as their spatiotemporal interplay, and the microenvironment conformation where they drive HO in FOP.

Schoenmaker and colleagues have shown that activin A promotes the development of fewer, albeit more expanded and active, osteoclasts, independent of the FOP mutation, resulting in osteoclastogenesis induction of unknown underlying mechanism [105]. More recently, they have demonstrated that the presence of activin A instigates transcriptional changes in osteoclast formation both in healthy individuals and patients with FOP, likely through the upregulation of genes involved in the differentiation and function of osteoclasts, cell fusion, and inflammation [106]. However, Ye et al. reported no significant differences in serum levels of activin A, BMP4 or BMP6 between healthy individuals and FOP patients, nor between FOP patients with active flare-ups or in remission, suggesting that activin A, BMP4 and BMP6 may be instigators for flare-ups in FOP, but not biomarkers for FOP disease activity [107]. This finding is consistent with previous reports that indicate that, although activin A intensifies dysregulated BMP signaling in human FOP primary CTPCs via stimulation of the pSmad1/5/8 pathway to induce chondro-osseous differentiation, basal overstimulated pSmad1/5/8 signaling in FOP can be independent of activin A and BMP4 [108].

Despite post-injury activin A production in both traumatic HO (tHO) and FOP HO, single cell RNA sequencing (scRNA-seq) data reveal that activin A is produced by different cell types, suggesting that the precursor signals for HO formation may be distinct between FOP and traumatic HO [104]. Activin A inhibition can effectively block HO in FOP, highlighting the importance of the pro-osteogenic signal induced and sustained by activin A as a critical antecedent event in genetic HO [104]. Findings stemming from the phase 2 LUMINA-1 clinical trial using garetosmab, an activin A-blocking antibody, showed suppressed development of new HO lesions, as well as a lower volume of new abnormal bone formation lesions [109]. In contrast, antibodies targeting ACVR1 activation by its ligands have been shown to stimulate HO and activate the signaling of FOP-mutant ACVR1 in some circumstances [110].

Additionally, a newly developed adeno-associated virus (AAV)-based gene therapy approach that carries the combination of a codon-optimized human ACVR1 and engineered miRNAs targeting activin A and its receptor ACVR1^R206H^ has been demonstrated to be effective for downregulating the BMP-Smad1/5 signaling pathway and the osteogenic differentiation of heterozygous *ACVR1^R206H/+^* skeletal progenitors, leading to protection against trauma-induced HO in FOP mice [111]. Another AAV-compatible artificial miRNA gene therapy strategy, ablating the activin A signal and suppressing chondrogenic and osteogenic differentiation, has been proven efficacious in the prevention of HO in FOP mice [112]. Interestingly, constitutive overexpression of WT ACVR1 in FOP mice has been shown to rescue the murine perinatal lethality associated with the disease and inhibit spontaneous abnormal bone formation and injury-induced HO in FOP mice [113]. Considering the highly variable nature of FOP disease presentation, and the fact that genetic deficiency of the WT *Acvr1* allele aggravates HO, it has been recently suggested that the ratio of WT and mutant receptors may impact FOP severity [113]. It has also been proposed that overexpression of WT *ACVR1* may protect against abnormal skeletogenesis by increasing the levels of activin A-bound non-signaling complexes (NSCs) with WT ACVR1, thus decreasing osteogenic signaling in response to activin A [113]. Further corroboration of the placating effects of activin A-bound NSCs on FOP HO can be found from the finding that ‘agonist-only’ activin A muteins that activate ACVR1B but are unable to form NSCs with WT ACVR1 lead to more aggravated disease pathology, highlighting the finger two tip loop (F2TL) region of activin A as a crucial component of the NSC formation [114].

## 10. Inflammation as a Therapeutic Target in Fibrodysplasia Ossificans Progressiva

Because inflammation is a major contributor to FOP and other types of HO, a major avenue for treating HO has focused on disrupting the inflammatory drivers that lead to abnormal bone formation (Figure 2). The mainstay for FOP therapy remains glucocorticoids, such as prednisone, in order to decrease the inflammation in an FOP flare [28]. However, detailed study of the benefits of steroids remains sparse. Retinoid signaling is a key regulator of chondrogenesis. Downregulation is required for HO formation in order to trigger the endochondral ossification pathway [115]. Therefore, retinoid agonists hold potential as a therapy by which to block chondrogenesis [116]. Palovarotene is a potent retinoic acid receptor gamma (RARγ) agonist that can reduce HO in FOP mouse models [117,118] and in a blast-related HO rat model [119]. Palovarotene dampened systemic and local inflammatory responses and specifically reduced levels of IL-6, TNF-α, and IFN-γ [119]. Palovarotene has also been shown to downregulate the inflammatory microenvironment and decrease HO formation of tendon stem cells [120]. Clinical trials for the treatment of FOP patients using palovarotene (NCT03312634, NCT02190747, and NCT02279095) showed 50–60% efficacy in reducing new HO formation [121], although detailed analyses of the immune response in patients receiving palovarotene have yet to be undertaken. In addition, direct inhibition of activin A by the neutralizing antibody garetosmab is being tested in clinical trials as a potential target for blocking HO in FOP (NCT03188666) and has shown efficacy in decreasing the flare activity in patients with FOP [109].

Several other strategies that are commonly tried in patients with non-genetic HO have also been tried in patients with FOP. Rapamycin affects the mTOR signaling pathway. The mTOR pathway is recognized as a critical driver of hypoxia and inflammation—two factors that occur in traumatic injuries [122] and in FOP [102,123]. Rapamycin can prevent HO after blast-related injury in rats; this is thought to occur via the suppression of the expression of inflammatory genes like *Cxcl5*, C-X-C motif chemokine ligand 10 (*Cxcl10*), *IL-6*, and *Ccl2* [124]. This background supports the testing of rapamycin in clinical trials as a potential therapy for FOP [123,125,126].

The NSAID celecoxib is a potent cyclooxygenase-2 inhibitor that appears to significantly reduce HO formation after surgical trauma in mouse models [127]. Clinical studies have shown some positive results for patients with non-genetic HO [128]. Surprisingly, other types of NSAIDs do not appear to have similar reductions in HO formation; for example, indomethacin was seen to fail to prevent HO in a blast-related HO rat model [129]. More research is needed to better understand how and which NSAIDs may benefit patients with FOP and how these responses differ from non-genetic HO.

Targeting inflammatory pathways directly has shown some promise for FOP and other forms of HO. A case report of anti-IL1 therapy with anakinra or canakinumab suggests a potential benefit for patients with severe intractable FOP disease activity [103]. IL1 may also have a more general role, as it has also been implicated in a mouse model of neurogenic HO [130]. Targeting the Janus kinase 1 and 2 (JAK1/2) tyrosine kinase is another strategy for reducing inflammation. One factor produced by the pro-inflammatory cytokine OSM, which is normally produced by myeloid cells [76], is phosphorylation of signal transducer and activator of transcription 3 (Stat3). Stat3 is a part of the JAK-STAT signaling pathway and is a critical regulator of cytokine production. In a neurogenic HO mouse model induced by spinal cord injury, ruxolitinib reduced phosphorylation of Stat3 and decreased the formation of new HO [76]. Recently, the JAK inhibitor tofacitinib was reported to decrease inflammatory flare-ups in a cohort of patients with FOP [131]. Saracatinib, which has shown anti-inflammatory activity in atherosclerosis [132], is being investigated as a possible therapy for FOP [133]. Imatinib, another kinase inhibitor with anti-fibrotic and anti-inflammatory activity [134,135,136], has also been described for use in severe cases of FOP [137,138]. While most of these strategies are off-label and do not yet have full studies showing an ability to reduce new HO formation, they provide an important foundation upon which to study the role of inflammatory mediators in the prevention of new HO.

## 11. Future Research Directions and Conclusions

Inflammation and trauma are key factors influencing the development of HO, yet the precise mechanisms connecting these processes, as well as potential therapeutic targets within, largely remain a mystery. It is clear that these mechanisms involve macrophages, mast cells, and adaptive immune cells, each playing distinct roles throughout the initiation and progression of HO (Table 1). There are shared characteristics between inflammation in FOP and other types of HO that provide both translational relevance for understanding the inflammatory drivers and also for finding therapies that may be effective for both forms of HO. These include the engagement of hypoxia signaling pathways, the initiation of MSC differentiation, the targeting of inflammatory pathways such as IL1, and the activation of vascular signaling pathways.

There are still many critical questions yet to be answered about HO formation. For instance, some rheumatic diseases, such as systemic scleroderma, have been reported to be complicated by HO [139]. However, we still do not understand why these inflammatory disorders result in HO, yet other chronic inflammatory and rheumatic diseases result in bone loss [7]. These cellular interactions between the inflammatory triggers and tissue resident skeletal stem cell lineages remain an area of active research.

Pharmacologic modulation of the immune system appears to be a promising strategy for the mitigation of HO. However, our understanding of exactly how pharmacologic modulation of immunity reduces (or potentially enhances) HO formation remains largely empirical. Thus, a systematic understanding of the immune process is necessary for us to develop effective treatment strategies that likely combine targets in multiple pathways. Continued research into these mechanisms holds high potential for new knowledge that will guide future treatments to prevent and reverse HO.

## Figures and Tables

**Figure 1 biomolecules-14-00357-f001:**
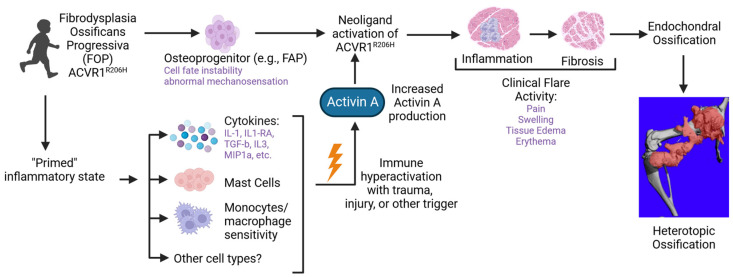
Potential inflammatory contributors to the pathogenesis of FOP. The *ACVR1^R206H^* genetic variant causes osteoprogenitors, such as fibroadipocyte progenitor cells (FAPs) to have an increased ability to enter the osteogenic pathway, possibly due to cell fate instability [49] and the abnormal mechanosensation [50] seen in other cell types. The *ACVR1^R206H^* variant also causes innate immune cells to be in a “primed” inflammatory state [20], where a subsequent trigger results in immune hyperactivation. Macrophages appear to be a potential source of activin A [51], which then leads to neo-ligand activation of ACVR1^R206H^ [52], with subsequent tissue inflammation, fibrosis, and endochondral ossification leading to the formation of heterotopic ossification. Figure created with Biorender.

**Figure 2 biomolecules-14-00357-f002:**
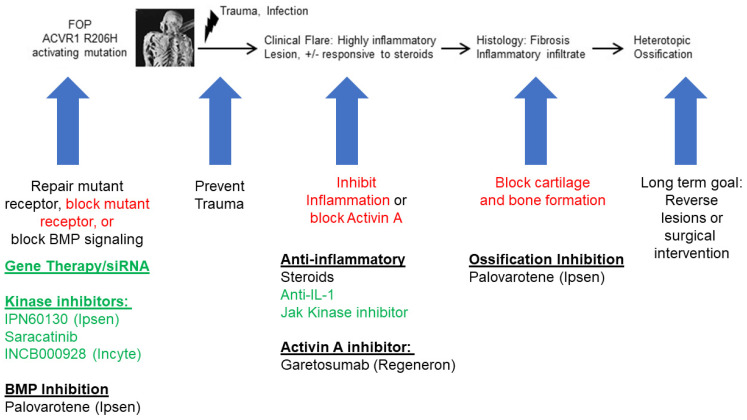
Potential therapies and investigational compounds for FOP and their relation to inflammation. Multiple therapies are being used or developed to treat FOP. These include investigational agents (green), and current medications used for standard of care. Palovarotene is approved by the US Food and Drug Administration and by Health Canada. Therapies that directly target the ACVR1 receptor or signaling pathway will also likely mitigate the inflammatory response induced by the FOP mutation. The long-term goal is to eventually develop strategies by which to reverse or allow for surgical resection of HO in patients with FOP.

**Table 1 biomolecules-14-00357-t001:** High-level summary of immune cells, including macrophage, mast cells, and adaptive immune cells, and related molecular target and intervention for FOP.

Immune Cell Type		Secreted Cytokines		Molecular Targets		Involvement in FOP		Intervention Strategies		References
Macrophages		IL-1β, IL-6, TGF-β, TNF-α, BMPs, activin A, OSM, SP, NT-3		BMP signaling pathway		Elevated levels of pro-inflammatory cytokines observed in FOP lesions, contributing to inflammation and aberrant tissue repair.		Targeting BMP signaling with inhibitors. Mitigating inflammation and aberrant bone formation using anti-inflammatory medications.		[16,19,62,63]
Mast Cells		IL-6, IL-9, TNF-α		Histamine receptors		Increased mast cell infiltration observed in FOP lesion, potentially contributing to inflammatory responses and tissue remodeling.		Blocking histamine receptors with antagonists to reduce mast cell activation and inflammation.		[20,62,86,87]
Adaptive Immune Cells (T cell and B cell)		IL-2, IL-4, IL-10, IL-17, IFN-γ		T cell receptors, B cell receptors		T cells may modulate inflammatory responses in FOP through cytokine production and regulation of immune cell activity. Cytokine production by B cells may contribute to inflammation and immune dysregulation.		Immune suppressive therapies targeting T and B cell activity to reduce inflammation and tissue damage.		[88,89,90]

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
