# Peer review of "Immunologic Aspects in Fibrodysplasia Ossificans Progressiva"

_biomolecules, 2024, doi:10.3390/biom14030357_

Round 1

Reviewer 1 Report

Comments and Suggestions for Authors

This is a very nice review about the immune regulation of FOP. It is very well written, which provides an excellent overview about the relationship of injury, inflammation, and excessive and abnormal bone formation during HO pathogenesis. My only suggestion is to have a Table or a schematic diagram to summarize the role of immune cells including macrophage, mast cells, and adaptive immune cells, and related molecular target and intervention for HO/FOP.    

Reviewer 2 Report

Comments and Suggestions for Authors

SUMMARY and COMMENTS

The review entitled “Immunologic Aspects in Fibrodysplasia Ossificans Progressiva” provides an overview of the implication of inflammation and the immune system in the context of Immunologic and inflamamtory Aspects in Fibrodysplasia Ossificans Progressiva, which is rare genetic disorder relying on the abnormal bone formation in tissues that are not normally mineralized, such as skeletal muscle, tendons and connective tissues. An important section on the potential therapeutic strategies aimed in targeting the impaired immune system and inflammatory players is provided, too. This is a nicely written and well-organized review. Considering the rarity of the disease the topic is not well debated in the literature, so the review manuscript presents an adequate novelty for the readers. The topic is adequately covered in detail and discussed. The two figures are well designed and informative. The manuscript requires several improvents and modifications. Below my comments and observations

Major concerns: 

1. Unless differently indicated by the journal guidelines, I suggest improving the abstract by providing background, rationale, purpose of the review and future perspectives. 

2. Additional two introductive sections briefly describing the main aspects of 1) Inflammatory responses and inflammation-associated molecules and immune cells 2) molecular aspects of the osteogenic and chondrogenic differentiation, should be included. Numerous molecules and factors are described in the context of Fibrodysplasia Ossificans Progressiva without, however, an appropriate introduction. These novel sections would drive the reader to the main topic

3. Some important literature is missing. More recently published literature on the role of inflammation immune system and bone formation in general should be included:  PMID: 34790042, PMID: 35163708, https://www.frontiersin.org/journals/immunology/articles/10.3389/fimmu.2020.00058/full  , https://www.frontiersin.org/journals/physiology/articles/10.3389/fphys.2020.511799/full 

https://www.sciencedirect.com/science/article/abs/pii/S0020138323009208 

https://inflammregen.biomedcentral.com/articles/10.1186/s41232-023-00279-1 

https://www.mdpi.com/1422-0067/20/20/5154 

Minor observation:

4. Please check the work for the presence of typos 

5. Please avoid abbreviations in the subhead titles (e.g., line 63).

6. The following supporting references (PMID: 32998835 and PMID: 31795299) on the role of cytokines in inflammation should be included in lines 49-51

7. A table summarizing the main molecules described in the review should be included. 

Comments on the Quality of English Language

English Language is fine
